# Variations of the Locomotor Profile, Sprinting, Change-of-Direction, and Jumping Performances in Youth Soccer Players: Interactions between Playing Positions and Age-Groups

**DOI:** 10.3390/ijerph19020998

**Published:** 2022-01-17

**Authors:** Ana Filipa Silva, Sümer Alvurdu, Zeki Akyildiz, Georgian Badicu, Gianpiero Greco, Filipe Manuel Clemente

**Affiliations:** 1Escola Superior Desporto e Lazer, Instituto Politécnico de Viana do Castelo, Rua Escola Industrial e Comercial de Nun’Álvares, 4900-347 Viana do Castelo, Portugal; anafilsilva@gmail.com (A.F.S.); filipe.clemente5@gmail.com (F.M.C.); 2The Research Centre in Sports Sciences, Health Sciences and Human Development (CIDESD), 5001-801 Vila Real, Portugal; 3Research Center in Sports Performance, Recreation, Innovation and Technology (SPRINT), 4960-320 Melgaço, Portugal; 4Faculty of Sport Sciences, Gazi University, Ankara 06500, Turkey; sumeralvurdu@gazi.edu.tr (S.A.); zekiakyldz@hotmail.com (Z.A.); 5Department of Physical Education and Special Motricity, Faculty of Physical Education and Mountain Sports, Transilvania University of Braşov, 500068 Braşov, Romania; georgian.badicu@unitbv.ro; 6Department of Basic Medical Sciences, Neuroscience and Sense Organs, University of Study of Bari, 70121 Bari, Italy; 7Instituto de Telecomunicações, Delegação da Covilhã, 1049-001 Lisboa, Portugal

**Keywords:** football, athletic performance, physical fitness, exercise test

## Abstract

The purpose of this study was two-fold: (i) analyze the variations of locomotor profile, sprinting, change-of-direction (COD) and jumping performances between different youth age-groups; and (ii) test the interaction effect of athletic performance with playing positions. A cross-sectional study design was followed. A total of 124 youth soccer players from five age-groups were analyzed once in a time. Players were classified based on their typical playing position. The following measures were obtained: (i) body composition (fat mass); (ii) jump height (measured in the countermovement jump; CMJ); (iii) sprinting time at 5-, 10-, 15-, 20-, 25- and 30-m; (iv) maximal sprint speed (measured in the best split time; MSS); (v) COD asymmetry index percentage); (vi) final velocity at 30-15 Intermittent Fitness Test (V_IFT_); and (vii) anaerobic speed reserve (ASR = MSS − V_IFT_). A two-way ANOVA was used for establishing the interactions between age-groups and playing positions. Significant differences were found between age-groups in CMJ (*p* < 0.001), 5-m (*p* < 0.001), 10-m (*p* < 0.001), 15-m (*p* < 0.001), 20-m (*p* < 0.001), 25-m (*p* < 0.001), 30-m (*p* < 0.001), V_IFT_ (*p* < 0.001), ASR (*p* = 0.003), MSS (*p* < 0.001), COD (*p* < 0.001). Regarding variations between playing positions no significant differences were found. In conclusion, it was found that the main factor influencing changes in physical fitness was the age group while playing positions had no influence on the variations in the assessed parameters. In particular, as older the age group, as better was in jumping, sprinting, COD, and locomotor profile.

## 1. Introduction

The soccer match requires from the players a well and multilateral developed physical fitness [1,2]. In fact, considering that the soccer game is an intermittent exercise with a mix of bioenergetic demands it is expectable to observe in players a good ability to sustain prolonged efforts and, at the same time, the availability to intensify the actions in more powerful movements [3,4]. Looking into the physical demands of the match, most of the time is spent in low-to-moderate running intensities [5,6], although the percentage of time spent in high intensity running or sprinting has been increasing over the years for the same total distance covered [7,8,9]. Thus, the match is becoming more intense requiring from the players a superior level of conditioning to sustain such intensifications [10,11]. Although ultimate performance in soccer is multifactorial, some of the critical events (e.g., goal-oriented events) are closely related to high-intensity running demands [5,6]. Thus, holding a good physical fitness can be a support for the ultimate performance [12,13]. 

Characterizing physical fitness is now a well-implemented practice in soccer [14]. The battery of tests is commonly used in different periods of the seasons helping strength and conditioning coaches to individualize the training process, identify the status of the players and observe the evolution trends of the players over the season [14]. Moreover, in the particular case of youth, longitudinal observations/assessments also allow to identify talents across the time. Although talent identification is a complex and multidimensional process [15,16], using physical fitness parameters as part of talent identification processes is still prevalent [17]. As example, using physical fitness batteries allows to identify that in some age groups the taller, heavier and more physical advanced players are those with higher levels [18,19]. Although these facts not being exclusively related to the ultimate players’ selection and transition for professional careers, tracking players over time can provide some additional information about which expectations coaches should have regarding their players and the evolution trends of the players over time and also determine how players can cope with match intensity [20].

For the case of physical fitness, it seems that the breaking point of 14/15 years old is the one in which change-of-direction (COD), linear sprint, standing long jump and aerobic capacity tests makes more sense are more sensitive to age-related changes in functional characteristics [21,22]. Moreover, testing batteries consisting in either vertical/horizontal jumps, sprinting and COD and aerobic fitness seems to be sensitive enough to distinguish between different youth age groups [23]. Interestingly, the most common tests as countermovement jump (CMJ), 5-0-5 (COD test), 10- to 20-m linear sprint test or standing broad jump are proven to be highly reliable and valid for youth soccer players [24]. 

Although the above-mentioned tests present a good consensus about the usability for practice, some other tests can be used directly helping coaches to prescribe the training process and classify the youth players. As example, the 30-15 Intermittent Fitness Test (30-15IFT) has been used for standardize the training intensity while applying high-intensity interval training [25]. Moreover, combining the final velocity at 30-15IFT and the maximal sprint speed (MSS) it is possible to obtain the anaerobic speed reserve (ASR) of the players and classify them into their locomotor profile (e.g., speed, hybrid, and endurance) [26].

Observing positive changes of physical fitness across the age-groups seems to be expectable [27]. However, in the context of soccer, playing positions seems to play an important role to differentiate players [27]. In a study conducted in a large sample of 744 youth players it was observed that after the age of 15, the attackers tends to be more explosive, the fastest and more agile players [28]. This tendency of observing greater differences in the later stages of development programs was also confirmed in a study conducted in 465 youth players [29].

The relevance of characterization the progression of physical fitness across age-groups, while in interaction with playing position seems to be obvious. This may help coaches to better specify and individualize the training process and classify the players based on their abilities to sustain and maintain match intensities. Therefore, the purpose of this study was two-fold: (i) analyze the variations of locomotor profile, sprinting, change-of-direction (COD) and jumping performances between different youth age-groups; and (ii) test the interaction effect of athletic performance with playing positions.

## 2. Materials and Methods

### 2.1. Study Design

This study followed a cross-sectional design. Players were recruited in the same team and no randomization was made. Age groups of 19 and 17 years old were assessed on 31 August 2021 and 1 September 2021. Age groups of 16 and 15 years old were assessed on 1 September 2021 and 2 September 2021. Age group of 14 years old was assessed on 2 September 2021 and 3 September 2021. The study begun after 3 weeks of the beginning of the season. As context, 24 number of training sessions were performed, and 3 friendly matches occurred before the study begun. 

### 2.2. Participants

The G*Power (version 3.1.) was used to calculate the a priori sample size. For a partial eta squared of 0.6 (medium effect size), a *p* = 0.05, power of 0.80, numerator df of 8 and number of groups of 10, the suggested total sample size was 20. A total of 124 young male elite male soccer players from U15 (*n* = 29), U16 (*n* = 30), U17 (*n* = 27), U18 (*n* = 25), and U19 (*n* = 12) teams were recruited voluntarily to participate in the study. All these players were regularly involved in two training sessions a week (90 min per session) with participation in one match at the weekend. Players and their guardians were informed about the study design and protocol. After being informed for potential risks of the study, guardians signed informed consent forms. This study followed the ethical principles of the Helsinki Declaration for human research. A local ethics committee also approved the protocol. Inclusion criteria for the participants were (i) being an active player with at least three years license, (ii) no history of any injuries during the previous two months, (iii) participating %85 of the training during the study period. 

### 2.3. Testing Procedures

The study were carried out in two different days, separated by a minimum of 48 h. On the first day, anthropometric assessment (height, body mass and body fat percentage), and performance tests (vertical jumping, sprinting and change-of-direction ability) were applied respectively. The assessments of the first day occurred at 2:00 p.m. of the day, in a room conditioned at 24 degrees Celsius and 52% relative humidity. Second day, 30-15 IFT test were performed to evaluate the final velocity (V_IFT_) and anaerobic speed reserve (ASR) in the following conditions: 03:00 p.m., 19 degrees Celsius and 49% relative humidity. Players were familiarized with all test at the previous seasons. All performance tests were conducted on a synthetic turf field (where the players train and compete) after a standardized FIFA 11+ warm-up protocol [30] (ref).

#### 2.3.1. Anthropometry

A measuring tape (SECA 206, Hamburg, Germany) and a digital scale (SECA 874, Hamburg, Germany) with an accuracy of 0.1 kg were used to measure the height and body mass of the participants. Body fat percentage was evaluated with 4-site skinfold measurement (biceps, triceps, iliac crest and subscapular) according to the Durnin–Womersley formula [31]. At least two measurements were taken from each athlete and if there was more than 5 percent difference between the two measurements, the third measurement was taken. 

#### 2.3.2. Jumping Performance 

Countermovement Jump (CMJ) was used to evaluate participants’ jumping performances with Optojump optical measurement system (OptojumpNext, Microgate, Bolzano, Italy). The participants performed three vertical attempts with 2 min recovery and the best attempt was used for the analyses. During the attempt, the participants were asked to jump keeping their hands on the hips and without bending the legs from take-off and landing phase. 

#### 2.3.3. Sprinting 

The 30 m linear sprint test with 5 m splits (5-, 10-, 15-, 20-, 25-, 30) were measured using the electronic timing gates system (Smartspeed, Fusion Sport, QLD, Australia). The timing gates were positioned at 1.2 m height of the floor. Players positioned 0.5 m far from the first timing gate and were encouraged to sprint at maximum speed and were given to two attempts with three minutes of recovery to prevent fatigue. Players took their preferred foot one step forward before the start and no signal was given. They started in split position, and always with the same preferred leg. The best sprinting time (lower value) was used for the analysis.

#### 2.3.4. Maximal Speed Sprint

The MSS was estimated using the average time over the last 10- and 5-m splits of a 30-m sprint test. A previous study revealed that using both 10- and 5-m splits of a 30-m sprint test while using timing gates can be reliability and present a high level of agreement with the MSS estimated using a gold-standard radar gun [32]. 

#### 2.3.5. Change-of-Direction Ability

The Arrowhead agility test was used for the participants’ COD ability. Electronic timing gates system (Smartspeed, Fusion Sport, QLD, Australia) positioned at the start line with a height of 1.2 m starting from the floor. The participant positioned 0.50 m from the timing gate and sprinted from the start line to the middle marker (A), turned to the left or right side to sprint around the marker (B), sprinted around the top marker (C) and sprinted back through the timing gate to finish the test [33]. Athletes were asked to use their right leg when they turned left, and their left leg when they turned right as breaking legs. The test was performed for left and right sides with four randomized attempts separated by at least three minutes of recovery. The best attempts of each side was recorded for analysis.

The asymmetry index was calculated according to the following formula [34]:Asymmetry Index percentage (AI%):  AI% = [(COD time _Dominant_ − COD time _Non-dominant_)/COD time _Dominant_] × 100

#### 2.3.6. Velocity at 30-15 IFT and Anaerobic Speed Reserve 

The 30-15IFT was performed by the participants according to the protocol developed by Buchheit [25]. The tests consist in perform 30 s shuttle runs interspersed with 15 s of passive recovery. The test starts with a velocity of 8 km/h. The speed increases by 0.5 km/h after each stage (30-s). Every time the player was unable to reach the line with the pace imposed by the audio beep, was marked. After failing three consecutive times, the final velocity achieved correctly was considered for further analysis. The last completed stage was used to determine the final velocity (V_IFT_) and anaerobic speed reserve was calculated as the difference between MSS and V_IFT_ with the following equation [35]: ASR (km/h) = MSS − V_IFT_

### 2.4. Statistical Analyses

Shapiro-Wilk and Levene tests were used to test the assumption of normality and homoscedasticity, respectively. Both, normality and homogeneity were confirmed with *p* > 0.05. Then, Bonferroni homoscedasticity and Two-way ANOVA were used, respectively. The Two Way ANOVA with Bonferroni post hoc test was used to compare player positions and ages. All statistical analyses were performed using RStudio Version: 2021.09.1 + 372. Statistics at a significance level of *p* < 0.05. The following scale was used to classify the effect sizes (ES) of the tests: small, 0.2–0.49; moderate, 0.50–0.79; large, 0.80–1. Partial eta-squared was used ANOVA and Cohen D to pairwise comparisons. 

## 3. Results

Two-way ANOVA tested interactions between age-groups and playing positions. No significant interactions were found on height (*p* = 0.031; η^2^_p_ = 0.235), body mass (*p* = 0.235; η^2^_p_ = 0.171), body fat (*p* = 0.635; η^2^_p_ = 0.121), CMJ (*p* = 0.027; η^2^_p_ = 0.239), 5-m (*p* = 0.412; η^2^_p_ = 0.146), 10-m (*p* = 0.490; η^2^_p_ = 0.137), 15-m (*p* = 0.582; η^2^_p_ = 0.127), 20-m (*p* = 0.464; η^2^_p_ = 0.140), 25-m (*p* = 0.178; η^2^_p_ = 0.182) and 30-m (*p* = 0.252; η^2^_p_ = 0.168), MSS (*p* = 0.388; η^2^_p_ = 0.149), VITF (*p* = 0.166; η^2^_p_ = 0.18 4), ASR (*p* = 0.441; η^2^_p_ = 0.143), COD right (*p* = 0.159; η^2^_p_ = 0.186), COD left (*p* = 0.662; η^2^_p_ = 0.118), and COD-AI% (*p* = 0.598; η^2^_p_ = 0.125).

One-way ANOVA tested the variations of physical fitness measures between age-groups. Descriptive statistics can be found in the Table 1 (anthropometrics) and Table 2 (physical fitness). Results revealed that the age group of 14 years old was significantly smaller and lighter (*p* < 0.05) than the remaining age groups. No other significant differences were found regarding anthropometric outcomes.

Results from Table 2 revealed that the younger age group (under-14) had significant smaller values of CMJ (*p* < 0.05), was significantly slower at 5-, 10-, 15-, 20-, 25- and 30-m distances and COD right (*p* < 0.05), and had significant smaller MSS, VIFT, and ASR (*p* < 0.05) than the remaining age-groups.

One-way ANOVA tested the variations of physical fitness measures between playing positions. Descriptive statistics can be found in the Table 3 (anthropometrics) and Table 4 (physical fitness. Results from Table 3 revealed that central defenders and forwards were significantly taller and heavier (*p* < 0.05) than the remaining positions. No significant differences were found regarding body fat.

Results from Table 4 revealed no significant differences between playing positions regarding the physical fitness outcomes.

In Figure 1, descriptive plots for anthropometry, CMJ, 5-m and 10-m were presented. Although no significant differences between playing positions, it is evident a significant difference of the younger age-group for being smaller and lighter than the remaining age groups. 

In Figure 2, descriptive plots for 15 m, 20 m, 25 m, 30 m, Maximum Speed, and ASR were presented. It seems evident a significant trend for being faster as older as players are (independently of the distance considered in the sprint test). Moreover, maximal speed sprint and anaerobic speed reserve also increase as players are older.

In Figure 3, descriptive plots for COD Right, COD Left, Asymmetry Index, COD-AI%, and VIFT were presented. As older players are, the better COD performance they get. Although no significant differences can be observed in the asymmetry index with exception of the pair of 15 and 16 years old. The VIFT is also significantly rising with the increase of age groups.

## 4. Discussion

The current cross-sectional study conducted over 124 youth soccer players revealed that age groups play a significant effect on physical fitness while playing positions were not capable to determine variations in physical fitness. Considering that significant changes in physical fitness were found between age groups, it was also observed that the older the groups, the better the results. Therefore, from 14 to 18 years old, the players turn taller, heavier, faster, while jumping higher and having a greater locomotor profile to sustain the efforts.

### 4.1. Age Group Comparisons

The normal growth patterns were found in the current research, namely considering the progressive increase of height and body mass until the last stage of youth [36,37]. Thus, the older the player is in youth soccer, the taller and heavier is. Such an evidence is confirmed in previous studies comparing different age groups withing the period of growth [38,39]. Interestingly, in the contrary to a possible expectation of observing an improvement in body fat levels [39,40], no significant differences were found across the age-groups in the current study. One of the causes could be the small body fat levels observed in the current study (mean values were stable around 8% over the ages) which is low, mainly in comparison to the studies reporting body fat in youth soccer players which presented values between 7 and 11% [36]. Also, in the opposite to expected [28,41], no significant differences were found in anthropometric and body composition data between playing positions. Although the current sample does not include goalkeepers (which is one of the positions favorable to be taller than remaining) [42], it would be expectable to observe significant variations between the remaining positions. As average (since interactions with age was not found), playing positions varied from 170 to 180 cm, while body mass between 60 and 70 kg. Although variations were observed, no significances were found, which may indicate that the tendency for selecting players based on playing position may not be too much implemented in the context of this group of players (considering that all of them belong to the same club).

In the current study it was found that as older the players as faster they are. Considering the different measures related with sprinting performance (e.g., 5-, 10-, 15-, 20-, 25-, 30-, and MSS) and COD performance it was observed a progressive and significant improvement until reach the final stage of youth soccer (i.e., 18 years old). These tendencies are in line with previous reports for youth soccer players [43,44]. Some possible explanations can be related with the growth and maturation that plays an important role in the muscular adaptation and neural drive, and bioenergetics to sustain MSS in late puberty [45]. Lower limb power observed in the improvements of CMJ over the age groups considered can possibly explain those advantages in neural and muscular adaptations over age. Although huge differences in the determinants that explains different linear sprint distances and COD, it was interesting to observe that older were always better in any of sprint test distances, COD measures and CMJ. 

Therefore, it can be argued that older tends to accelerate better (possibly explained by the greater concentric force and power which was possible observed by the increases in CMJ performance over the age-groups) [46], achieve higher velocities (possible explained by a greater eccentric force, vertical force and power) [47] and can decelerate and accelerate better due to the better neuromuscular properties [48] developed in accordance to the training process, and normal increase in muscular adaptation and neural drive. In, fact, considering that older can reach a greater MSS than younger [49], it is expectable that such a stimulus in match and training can play an important role in the development of sprinting and COD performance since achieving peak speed is an effective way of improve it [50,51]. However, as major factors can be listed maturation and the related neural function, multi-joint coordination, muscle stiffness, and changes in muscle architecture [52].

In the current study it was also found that locomotor profile determined by ASR and VIFT followed the trend of the older, the better. Considering that locomotor profile is highly associated with aerobic fitness, it is expectable to observe natural and progressive increases after the maturational peak until reach the 16 years old in males [52,53]. These changes and increases are potentially explained by changes occurring in central mechanisms namely considering the increase of heart, lungs, muscles and blood volume [54,55]. Naturally, other factors as hormonal or enzymatic can be also important for ultimately improving the progressive improvement of aerobic fitness during the youth stages [56]. Thus, this can justify improvements in aerobic power as well as in the maximal aerobic speed which may justifies improvements in V_IFT_ [57]. Considering that VIFT is justified by different measures including aerobic fitness, change-of-direction or lower limb power, and taking in consideration that the older, the better in these levels, VIFT tends to be improved also across the age-groups. Moreover, considering that anaerobic systems is improved after peak maturation [58,59], it is also expectable to assist to an improvement of ASR as well [60].

### 4.2. Playing Position Comparisons

One of the common trends observed over the current results were the absence of playing position effect on the physical fitness variation of youth soccer players. This is not in line with most of studies conducted in soccer, mainly those conducted in later stages of youth formation [28,29]. Possibly, a better fitness level observed can mask positional differences that traditionally occur in players based on the specificity of the training process and match demands. Future research should focus in analyze if a proper training process can mitigate differences between playing positions, or on the other side, a training process based on the average and not individuality can also decrease differences between playing positions.

### 4.3. Study Limitations, Future Research and Practical Implications

The fact of the study has been conducted in only one club can be a source of bias, like many other cross-sectional studies conducted in this field of research. Observational analytic studies to determine differences between age groups and playing positions should be made in the future with more than one context and determine how the context can play a role or not in the evidence collected. Despite the limitation, this study was conducted in 124 players which is substantial and allows a sample enough to confirm the evidence. As practical implications, this study may suggest that as older, as better. Thus, with the progression in age, a more focused stimulus can be provided on the physical fitness, and possible more individualization and specificity of training can occur to ensure the adjustment to the position specificities of the game.

## 5. Conclusions

This study revealed that the older, the better in terms of physical fitness in youth soccer players. Considering the age-groups included (14 to 18 years old), improvements in locomotor profile, sprinting, change-of-direction, and jumping performance were significant and obvious. Younger players were significantly smaller and lighter, while were significantly slower, jump smaller and had less maximal speed sprint, anaerobic speed reserve and V_IFT_. Although this evidence was not found significant interactions of age-group with playing positions and, additionally, playing positions did not differentiate athletes.

## Figures and Tables

**Figure 1 ijerph-19-00998-f001:**
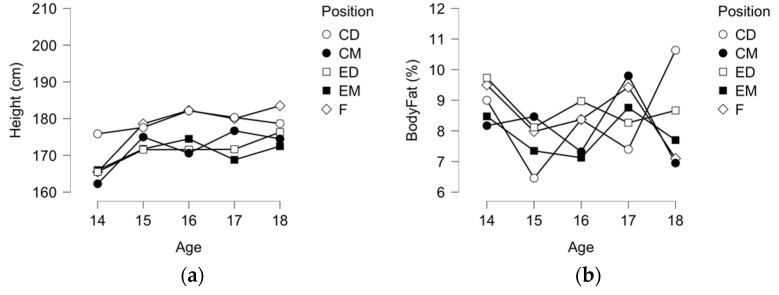
Descriptive plots for (a) height (cm); (**b**) body fat (%), (**c**) body mass (kg); and (**d**) CMJ (cm).

**Figure 2 ijerph-19-00998-f002:**
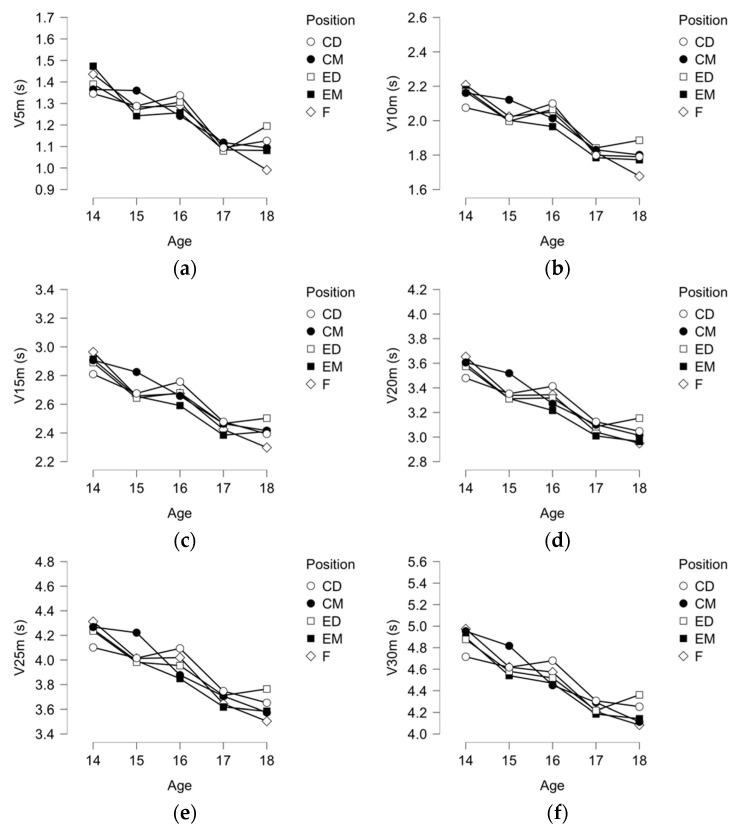
Descriptive plots for (**a**) 5-m; (**b**) 10-m; (**c**) 15-m; (**d**) 20-m; (**e**) 25-m; and (**f**) 30-m sprint time (s) and (**g**) maximal speed sprint (km/h); and (**h**) anaerobic speed reserve (km/h).

**Figure 3 ijerph-19-00998-f003:**
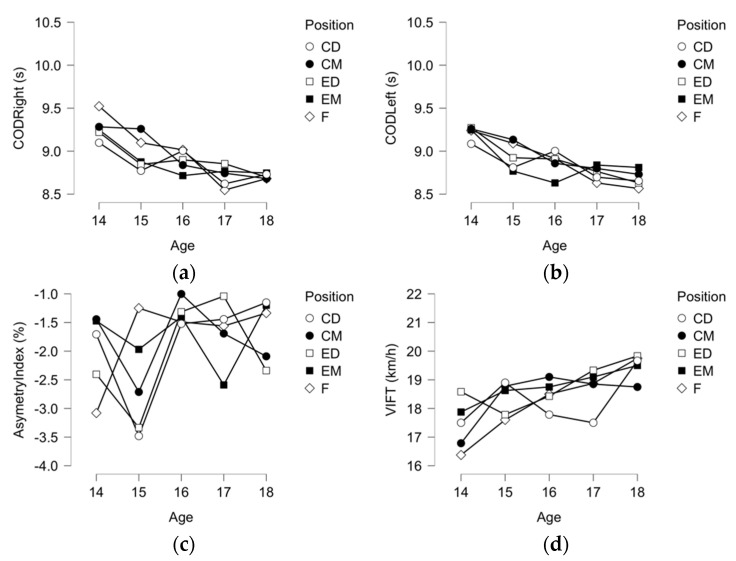
Descriptive plots (**a**) COD right leg (s); (**b**) COD left leg (s); (**c**) asymmetry index percentage (%); and (**d**) VIFT (km/h).

**Table 1 ijerph-19-00998-t001:** Descriptive statistics (mean and standard deviation) for the anthropometric outcomes between age-groups.

Measure	14 yo (N = 29)	15 yo (N = 30)	16 yo (N = 27)	17 yo (N = 25)	18 yo (N = 12)	*p*	ES
Height (cm)	167.65 ± 7.02 ^b,c,d,e^	174.80 ± 4.70 ^a^	176.14 ± 6.63 ^a^	175.48 ± 7.04 ^a^	177.16 ± 4.89 ^a^	0.001	0.243
BM (kg)	57.43 ± 7.90 ^b,c,d,e^	64.54 ± 5.79 ^a^	65.82 ± 5.96 ^a^	67.78 ± 6.52 ^a^	69.32 ± 5.77 ^a^	0.001	0.288
BF (kg)	8.94 ± 2.82	7.82 ± 1.69	8.14 ± 1.81	9.06 ± 2.30	8.45 ± 1.93	0.173	0.052

Yo: years old; BM: body mass; Body fat: BF; significant different from 14 yo ^a^; 15 yo ^b^; 16 yo ^c^; 17 yo ^d^; and 18 yo ^e^ at *p* < 0.05.

**Table 2 ijerph-19-00998-t002:** Descriptive statistics (mean and standard deviation) for the physical fitness outcomes between age-groups.

Measure	14 yo (N = 29)	15 yo (N = 30)	16 yo (N = 27)	17 yo (N = 25)	18 yo (N = 12)	*p*	ES
CMJ (cm)	36.95 ± 4.70 ^c,d^	37.96 ± 4.49 ^c^	42.08 ± 7.07 ^a,b^	42.02 ± 6.04 ^a^	41.04 ± 4.92	0.001	0.145
5-m (s)	1.39 ± 0.10 ^b,c,d,e^	1.29 ± 0.1 ^a,d,e^	1.29 ± 0.09 ^a,d,e^	1.10 ± 0.11 ^a,b,d,e^	1.10 ± 0.08 ^a,b,c^	0.001	0.517
10-m (s)	2.14 ± 0.11 ^b,c,d,e^	2.04 ± 0.11 ^a,d,e^	2.04 ± 0.09 ^a,d,e^	1.81 ± 0.14 ^a,b,c^	1.79 ± 0.10 ^a,b,c^	0.001	0.562
15-m (s)	2.88 ± 0.14 ^b,c,d,e^	2.70 ± 0.15 ^a,d,e^	2.68 ± 0.11 ^a,d,e^	2.44 ± 0.19 ^a,b,c^	2.41 ± 0.12 ^a,b,c^	0.001	0.566
20-m (s)	3.57 ± 0.16 ^b,c,d,e^	3.38 ± 0.16 ^a,d,e^	3.32 ± 0.14 ^a,d,e^	3.07 ± 0.20 ^a,b,c,e^	3.03 ± 0.12 ^a,b,c^	0.001	0.573
25-m (s)	4.21 ± 0.19 ^b,c,d,e^	4.06 ± 0.17 ^a,d,e^	3.97 ± 0.16 ^a,d,e^	3.68 ± 0.22 ^a,b,c^	3.63 ± 0.14 ^a,b,c^	0.001	0.571
30-m (s)	4.86 ± 0.22 ^b,c,d,e^	4.65 ± 0.20 ^a,d,e^	4.54 ± 0.18	4.24 ± 0.23 ^a,d,e^	4.21 ± 0.16 ^a,b,c^	0.001	0.570
MSS (km/h)	28.60 ± 1.69 ^b,c,d,e^	30.80 ± 2.72 ^a^	31.85 ± 2.75 ^a^	32.24 ± 2.21 ^a^	32.01 ± 1.88 ^a^	0.001	0.268
V_IFT_ (km/h)	17.44 ± 1.49 ^c,d,e^	18.35 ± 1.19	18.44 ± 1.08 ^a^	18.80 ± 1.24 ^a^	19.54 ± 0.89 ^a^	0.001	0.203
ASR (km/h)	11.15 ± 2.05 ^c,d^	12.45 ± 3.04	13.41 ± 2.80 ^a^	13.44 ± 2.28 ^a^	12.47 ± 2.30	0.007	0.111
COD right (s)	9.27 ± 0.25 ^b,c,d,e^	9.0 ± 0.36 ^a,d,e^	8.90 ± 0.29 ^a^	8.71 ± 0.22 ^a,b^	8.70 ± 0.14 ^a,b^	0.001	0.338
COD left (s)	9.20 ± 0.29	8.97 ± 0.31	8.88 ± 0.29	8.76 ± 0.25	8.67 ± 0.18	0.001	0.228
COD–AI%	−1.94 ± 1.19	−2.64 ± 2.50^c^	−1.34 ± 0.76 ^b^	−1.74 ± 0.89	−1.64 ± 1.35	0.029	0.086

Yo: years old; CMJ: countermovement jump; MSS: maximal sprint speed; VIFT: final velocity at 30-15 Intermittent fitness test; ASR: anaerobic speed reserve; COD: change-of-direction; COD-AI%: Change-of-Direction Asymmetry Index percentage; significant different from 14 yo ^a^; 15 yo ^b^; 16 yo ^c^; 17 yo ^d^; and 18 yo ^e^ at *p* < 0.05.

**Table 3 ijerph-19-00998-t003:** Descriptive statistics (mean and standard deviation) for the athropometric outcomes between playing positions.

Measure	CD (N = 26)	CM (N = 33)	ED (N = 26)	EM (N = 19)	F (N = 19)	*p*	ES
Height (cm)	178.73 ± 5.04 ^b,c,d^	172.12 ± 7.76 ^a,e^	170.73 ± 5.42 ^a,e^	170.42 ± 3.97 ^a,e^	177.42 ± 7.55 ^b,c,d^	0.001	0.235
BM (kg)	68.61 ± 4.61 ^b,c,d^	62.73 ± 8.85 ^a^	62.19 ± 7.42 ^a^	61.46 ± 5.03 ^a^	66.64 ± 8.36	0.002	0.132
BF (kg)	8.34 ± 2.30	8.54 ± 2.16	8.80 ± 2.44	7.94 ± 1.75	8.59 ± 2.30	0.772	0.015

CD: central defender; ED: external defender; CM: central midfielder; EM: external midfielder; F: forward; BM: body mass; Body fat: BF; significant different from CD ^a^; CM ^b^; ED ^c^; EM ^d^; and F ^e^ at *p* < 0.05

**Table 4 ijerph-19-00998-t004:** Descriptive statistics (mean and standard deviation) for the physical fitness outcomes between playing positions.

Measure	CD (N = 26)	CM (N = 33)	ED (N = 26)	EM (N = 19)	F (N = 19)	*p*	ES
CMJ (cm)	38.88 ± 5.76	38.68 ± 4.58	39.30 ± 5.81	42.31 ± 7.87	40.88 ± 5.72	0.195	0.050
5-m (s)	1.27 ± 0.14	1.25 ± 0.14	1.27 ± 0.13	1.23 ± 0.18	1.25 ± 0.16	0.861	0.011
10-m (s)	2 ± 0.15	2 ± 0.17	2.02 ± 0.14	1.95 ± 0.21	1.98 ± 0.21	0.714	0.018
15-m (s)	2.68 ± 0.20	2.68 ± 0.22	2.67 ± 0.19	2.59 ± 0.26	2.63 ± 0.25	0.688	0.019
20-m (s)	3.34 ± 0.21	3.34 ± 0.25	3.32 ± 0.21	3.23 ± 0.30	3.30 ± 0.27	0.594	0.023
25-m (s)	3.99 ± 0.23	3.98 ± 0.29	3.97 ± 0.23	3.87 ± 0.33	3.94 ± 0.31	0.665	0.020
30-m (s)	4.58 ± 0.25	4.58 ± 0.33	4.56 ± 0.28	4.46 ± 0.37	4.54 ± 0.35	0.699	0.018
MSS (km/h)	30.76 ± 2.50	30.52 ± 0.64	31.18 ± 2.98	31.34 ± 2.61	31.06 ± 2.87	0.820	0.013
V_IFT_ (km/h)	18.09 ± 1.34	18.42 ± 1.50	18.55 ± 1.13	18.71 ± 1.05	18.02 ± 1.64	0.411	0.033
ASR (km/h)	12.67 ± 2.57	12.10 ± 2.60	12.63 ± 3.16	12.63 ± 2.34	13.03 ± 2.66	0.806	0.013
COD right (s)	8.91 ± 0.36	9 ± 0.32	8.93 ± 0.34	8.87 ± 0.30	9 ± 0.38	0.591	0.023
COD left (s)	8.91 ± 0.31	8.99 ± 0.34	8.94 ± 0.35	8.86 ± 0.33	8.93 ± 0.32	0.741	0.016
COD–AI%	−1.90 ± 1.97	−1.83 ± 1.35	−2.19 ± 1.99	−1.82 ± 1.05	−1.76 ± 1.18	0.890	0.009

CD: central defender; ED: external defender; CM: central midfielder; EM: external midfielder; F: forward; BM: body mass; Body fat: BF; CMJ: countermovement jump; MSS: maximal sprint speed; VIFT: final velocity at 30-15 Intermittent fitness test; ASR: anaerobic speed reserve; COD: change-of-direction; COD-AI%: Change-of-Direction Asymmetry Index percentage; significant different from CD; CM; ED; EM; and F at *p* < 0.05.

## Data Availability

Raw data of this article are available upon request to corresponding author.

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
