# Peer review of "Variations of the Locomotor Profile, Sprinting, Change-of-Direction, and Jumping Performances in Youth Soccer Players: Interactions between Playing Positions and Age-Groups"

_ijerph, 2022, doi:10.3390/ijerph19020998_

Round 1

Reviewer 1 Report

The manuscript give adequate information about youth football player profile in according with age group and playing position. The content is clearly defined and justified the work; the material and methods chapter presents all variable and the design of the research; the cited literature is proper to content and in relation with aim of the study.

Suggestions:

1. In the results section the tables with results may be organized
separately for anthropometrics and physical fitness.
At every table and plots should be some comments of the results. 2. In the discussions section, I prorpose to introduce two subsections,
one in relation with age group and one in relation with playing position. 3. Conclusions may be more specific and detailed.

Reviewer 2 Report

I found this study very interesting. However, I have some suggestions that I would like to share with the Authors.

1) Study design should be described more precisely. Authors report that study was conducted between 31 Aug and 31 Nov and the measurements of each participant lasted two days. I suppose that training experience in the analyzed period was different for participants who were measured in September and different for those measured in November. The applied training loads and played matches may have influenced the results of fitness tests. Please explain this issue.

2) Please consider displaying only figures 1-3 without presenting tables 1-2. In my opinion figures 1-3 clearly show the data presented in both tables. In that case, the broader description of the figures would be recommended.

3) I found some editorial mistakes:

line 73. is: "you age groups", should be: "youth age groups"

line 278 is: "8 kg", should be: "8 %"

line 290 is "in the current", should be : "in the current research/study"
